# A phosphoproteomic approach reveals that PKD3 controls PKA-mediated glucose and tyrosine metabolism

Angel Loza-Valdes[1,2,*] , Alexander E Mayer[1,*] , Toufic Kassouf[2] , Jonathan Trujillo-Viera[1], Werner Schmitz[3] , Filip Dziaczkowski[2] , Michael Leitges[4], Andreas Schlosser[1], Grzegorz Sumara[1,2]

Members of the protein kinase D (PKD) family (PKD1, 2, and 3) integrate hormonal and nutritional inputs to regulate complex cellular metabolism. Despite the fact that a number of functions have been annotated to particular PKDs, their molecular targets are relatively poorly explored. PKD3 promotes insulin sensitivity and suppresses lipogenesis in the liver of animals fed a high-fat diet. However, its substrates are largely unknown. Here we applied proteomic approaches to determine PKD3 targets. We identified more than 300 putative targets of PKD3. Furthermore, biochemical analysis revealed that PKD3 regulates cAMP-dependent PKA activity, a master regulator of the hepatic response to glucagon and fasting. PKA regulates glucose, lipid, and amino acid metabolism in the liver, by targeting key enzymes in the respective processes. Among them the PKA targets phenylalanine hydroxylase (PAH) catalyzes the conversion of phenylalanine to tyrosine. Consistently, we showed that PKD3 is activated by glucagon and promotes glucose and tyrosine levels in hepatocytes. Therefore, our data indicate that PKD3 might play a role in the hepatic response to glucagon.

## Introduction

Protein kinase D (PKD) family members integrate multiple hormonal and metabolic signals to coordinate homeostasis of the organism (Sumara et al, 2009; Rozengurt, 2011; Löffler et al, 2018; Mayer et al, 2019; Kolczynska et al, 2020; Trujillo-Viera et al, 2021). The family of PKDs comprises three kinases: PKD1, PKD2, and PKD3 (Fu & Rubin, 2011; Rozengurt, 2011). PKDs share a basic structure composed of the cysteine-rich domain, essential for their affinity for their main activators phorbol esters, and DAG. The pleckstrin homology domain (PH) and the C-terminal region determine the catalytic activity (Rozengurt et al, 1997; Iglesias et al, 1998). PKD1 and PKD2 share the highest homology, whereas PKD3 kinase is the unique member of the family. PKD1 and PKD2 have been widely studied in different cellular processes such as trans-Golgi network dynamics, cell proliferation, and cell migration, adipocytes and enterocyte function, insulin secretion as well as regulation of innate and adaptive immune cells function (Sumara et al, 2009; Rozengurt, 2011; Gehart et al, 2012; Ittner et al, 2012; Goginashvili et al, 2015; Zhang et al, 2017; Löffler et al, 2018; Mayer et al, 2019; Kolczynska et al, 2020; Trujillo-Viera et al, 2021). PKD3 has been implicated in tumor progression and invasiveness in breast and gastric cancers, as well as hepatocellular carcinoma (Huck et al, 2014; Yang et al, 2017; Zhang et al, 2019). Furthermore, recent research has demonstrated that PKD3 regulates insulin sensitivity, lipid accumulation, and fibrogenesis in the liver (Mayer et al, 2019; Zhang et al, 2020). Thus, PKD3 plays a role in a wide range of cellular processes in both physiological and pathological conditions.

To date, only a few downstream targets of PKD3 have been identified. PKD3 phosphorylates G-protein–coupled receptor kinase–interacting protein 1 (GIT1) on serine 46 to regulate the localization of GIT1-paxillin complex and consequently cell shape and motility (Huck et al, 2012). Moreover, ectopic expression of a constitutive active form of PKD3 (PKD3ca) in TNBC (triple-negative breast cancer cells) leads to hyperphosphorylation of S6 Kinase 1 (S6K1), a downstream target of the mechanistic target of rapamycin complex 1 (mTORC1), which is an energy sensor in the cell and sustains cell proliferation (Laplante & Sabatini, 2012; Huck et al, 2014). PKD3 also phosphorylates p65 at serine 536, a critical step for the up-regulation of 6-phosphofructo-2-kinase/fructose-2,6-biphosphatase 3 (PFKFB3) and drives glycolysis in gastric cancer cells (Zhang et al, 2019). In addition, gain and loss of function studies suggest that PKD3 regulates the ERK1-MYC axis and promotes cell proliferation in cancer (Chen et al, 2008; Liu et al, 2019). Finally, in hepatocytes, PKD3 suppresses insulin-dependent a Ser/Thr Kinase (AKT) and mTORC1/2 activation, which results in peripheral glucose intolerance and suppression of hepatic lipid production (Mayer et al, 2019). Nevertheless, the PKD3 targets in the liver and other organs remain largely unexplored.

[1]Rudolf Virchow Center, Center for Integrative and Translational Bioimaging, University of Würzburg, Würzburg, Germany    [2]Nencki Institute of Experimental Biology, Polish Academy of Sciences, Warsaw, Poland    [3]Theodor Boveri Institute, Biocenter, University of Würzburg, Würzburg, Germany    [4]Tier 1, Canada Research Chair in Cell Signaling and Translational Medicine, Division of BioMedical Sciences/Faculty of Medicine, Craig L Dobbin Genetics Research Centre, Memorial University of Newfoundland, Health Science Centre, St. Johns, Canada

Correspondence: g.sumara@nencki.edu.pl
*Angel Loza-Valdes and Alexander E Mayer contributed equally to this work

The liver has a major role in the regulation of glucose, lipid, and AAs homeostasis by regulating the adaptation to nutrient availability. In the liver, AAs are used to synthesize proteins and precursors for different bioactive molecules. Moreover, ammonia, a by-product of protein catabolism, is disposed of as urea by the liver (Waterlow, 1999; Bröer & Bröer, 2017). Under certain physiological conditions such as fasting, the liver can use AAs to produce glucose or ketone bodies. This metabolic response is hormonally regulated by glucagon, which is released from the pancreatic α cells (Holst et al, 2017; Petersen et al, 2017). PKA holoenzyme, composed of two regulatory and two catalytic subunits, is a master regulator of hepatic glucose and amino acids metabolism. PKA drives such processes as gluconeogenesis (from some amino acids, glycerol, and lactate) and glycogenolysis to maintain glucose levels during fasting (London et al, 2020). This kinase also stimulates the activity of such enzymes as PAH which regulates tyrosine (Tyr) levels (Miranda et al, 2002).

Phenylalanine (Phe) is an essential AA in mammals, and its conversion into Tyr is crucial for the production of thyroid hormones and catecholamines. The conversion into tyrosine is tightly regulated by the enzyme phenylalanine hydroxylase (PAH), an enzyme that requires tetrahydrobiopterin ($BH_4$) as a cofactor, and molecular dioxygen as a substrate (Kaufman, 1958; Fitzpatrick, 1999). Mutations in PAH lead to phenylketonuria (PKU), an abnormal accumulation of Phe in peripheral tissues (Konecki & Lichter-Konecki, 1991; Scriver, 2007; Williams et al, 2008). Of note, the expression of PAH is restricted to the liver and kidney, major organs involved in AAs metabolism (Hsieh & Berry, 1979). PAH activity is regulated allosterically by high intracellular levels of Phe and hormonally by glucagon and insulin. By contrast, it was recently shown that oxygen concentrations might affect PAH activity in hepatocytes because of oxygen zonation (Kaufman, 1958; Donlon & Kaufman, 1978; Ying et al, 2010). Upon fasting, glucagon rewires liver metabolism and promotes AA catabolism. Glucagon leads to an increase in cAMP (cyclic adenosine monophosphate) and activates PKA, which phosphorylates PAH at serine 16 to promote its function and increase the rate of Phe to Tyr conversion (Miranda et al, 2002).

Here, we carried out a phosphoproteomic study to investigate phosphorylation events dependent on PKD3. We found more than 300 direct or indirect targets of PKD3, among them PAH. Consistently, in mice and primary hepatocytes overexpressing constitutive active form of PKD3 (PKD3ca), Tyr levels were elevated, whereas the deletion of PKD3 resulted in decreased conversion of Phe to Tyr. Moreover, we showed that glucagon signaling promotes PKD3 activation which is required for glucagon-induced Phe to Tyr conversion. However, our data indicate that PKD3 does not phosphorylate PAH directly, but promotes the activity of PKA which phosphorylates PAH. Consistently, with the PKA function in the liver, PKD3 also promotes glucagon-induced and fasting glucose levels in mice. Taken together, we have identified potential PKD3 substrates in hepatocytes and uncovered the function of PKD3 in the regulation of Phe and Tyr metabolism as well as glucose homeostasis upon fasting.

# Results

### Unraveling putative targets of PKD3 in hepatocytes using substrate motif-specific antibodies

Previous research has delineated the role of PKD3 in the regulation of hepatic glucose and lipid metabolism in mice fed a high-fat diet (Mayer et al, 2019). However, the phosphorylation targets of PKD3 in the liver remain elusive. To unravel the putative targets of PKD3 in the liver we have used primary hepatocytes derived from PKD3 knockout mice and transduced these cells with adenovirus to overexpress either EGFP or PKD3ca (Fig 1A). Subsequently, protein lysates were isolated and used for pull-down assays. For this, we used PKD substrate motif–specific antibodies. PKD kinases recognize the consensus AA motif sequence LxRxx[S*/T*] (where L: leucine, R: arginine, S: serine, T: threonine, and x: any AA) within their putative targets (Döppler et al, 2005; Franz-Wachtel et al, 2012). Importantly, the arginine (R) in position −3 in relation to the phospho-acceptor is essential, whereas leucine (L) in −5 position might be in some cases replaced by other amino acids, for example, valine (V) or isoleucine (I) (Döppler et al, 2005). At first, phospho-(Ser/Thr) PKD substrate LxRxx[S*/T*] antibody was used for immunoprecipitation to enrich proteins that have a phosphorylated PKD motif in lysates from primary hepatocytes deficient for PKD3 expressing either EGFP control or PKD3ca (Fig 1A). Overexpression of PKD3ca showed an increase in proteins with a PKD motif (Fig 1B). Proteins with phosphorylated PKD motif were immunoprecipitated and characterized by mass spectrometry. 84 proteins were significantly enriched (significance of 1 or 2) in lysates from PKD3ca hepatocytes compared with the EGFP ones (Fig 1C and D and Table S1). The protein with highest enrichment induced by PKD3ca was PKD3 itself, suggesting that PKD3 might be subjected to autophosphorylation. Although leucine in −5 position is frequently present in PKD motifs, other targets also have valine (V) or isoleucine (I) in the −5 position (Döppler et al, 2005). Therefore, to complement our approach, we performed a similar experiment using an antibody only partially specific for PKD motif Rxx[S*/T*]. Overexpression of PKD3ca led to an increase in proteins that have a phosphorylated Rxx[S*/T*] motif (Fig 2A). Subsequently, proteins were immunoprecipitated, identified, and quantified by mass spectrometry. 226 proteins were significantly enriched (significance of 1 or 2) in lysates from PKD3ca hepatocytes compared with the EGFP expressing ones (Fig 2B and C and Table S2). Of note, using an antibody against motif Rxx[S*/T*], we identified almost three times more putative targets of PKD3 compared with antibody against LxRxx[S*/T*] motif, suggesting that PKD3 can frequently phosphorylate imperfect consensus site.

### Comparative analysis of putative PKD3 targets identified using antibodies against LxRxx[S*/T*] and Rxx[S*/T*] motifs

Our mass spectrometry screening identified 84 and 226 proteins significantly enriched using antibodies against LxRxx[S*/T*] and Rxx[S*/T*], respectively (Fig 3A). Of note, 24 proteins were enriched in both screenings (Fig 3A and C). In silico analysis revealed that 55% of proteins identified using an antibody against the LxRxx[S*/T*] motif have at least one putative PKD consensus side resembling the sequence [L/V/I]xRxx[S*/T*]. Similarly, also 55% of proteins enriched from hepatocytes expressing PKD3ca using an antibody against Rxx[S*/T*] had at least one [L/V/I]xRxx[S*/T*] motif in their sequence (Fig 3B). 12 of the proteins which had in their sequence an [L/V/I]xRxx[S*/T*] motif were enriched using both antibodies, against Rxx[S*/T*] and LxRxx[S*/T*] (Fig 3B). In silico analysis also revealed that these 12 proteins have in total 30 putative PKD motifs. Further analysis of the 30 putative PKD motifs carried out in the phosphosite.org repository

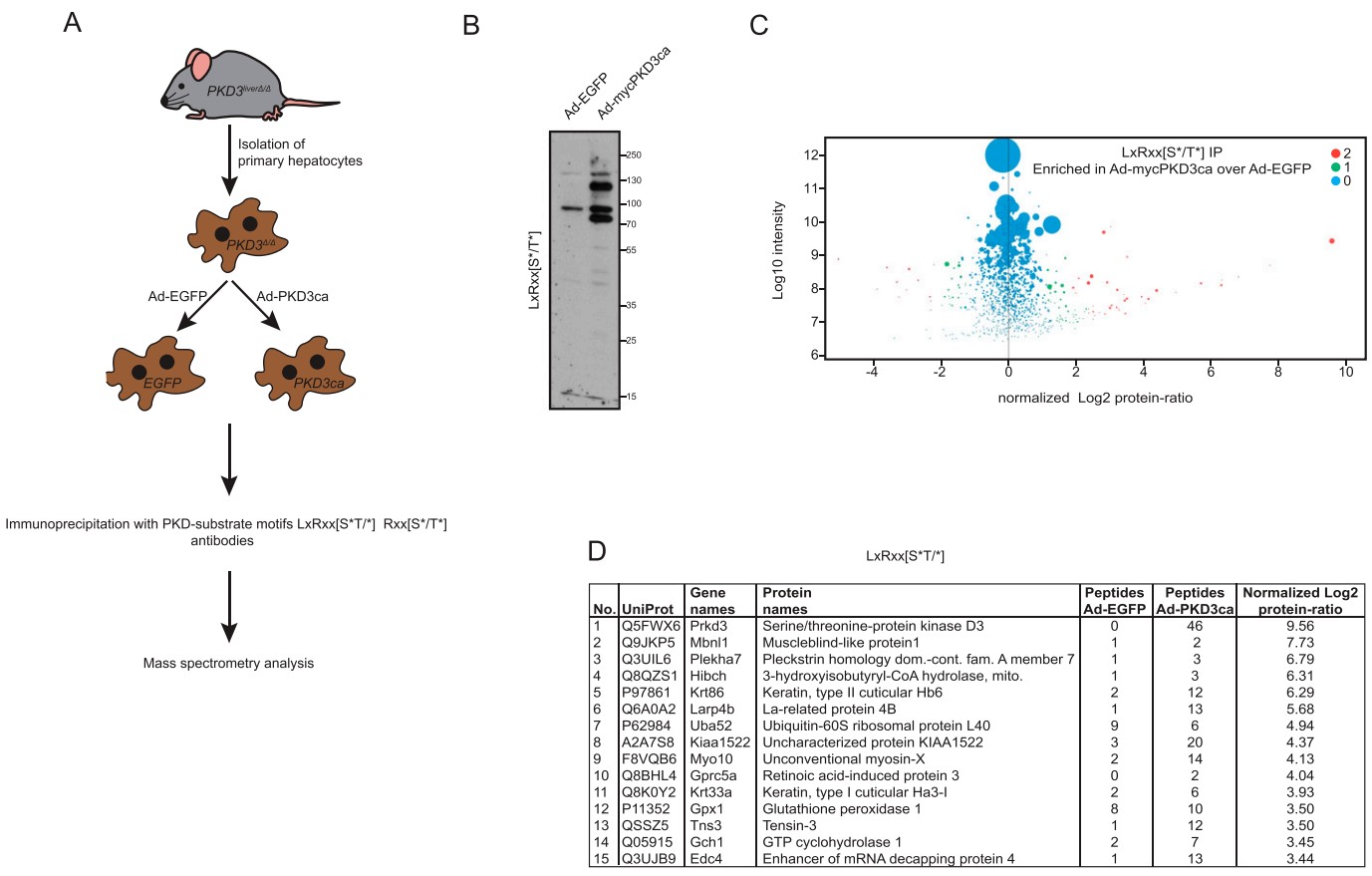

**Figure 1. Identification of protein kinase D (PKD)3 substrates by immunoprecipitation using PKD substrate motif antibody LxRxx[S*/T*].**
**(A)** Experimental design. Primary hepatocytes isolated from three PKD3-deficient mice were transduced by adenovirus containing EGFP (controls) or the constitutive active form of PKD3 (PKD3ca). IP with PKD-substrate motif antibodies was performed on cell extracts and followed by mass spectrometry analysis. **(B)** WB analysis of protein lysates from PKD3-deficient primary hepatocytes transduced with either adenovirus expressing control EGFP (Ad-EGFP) or constitutive active PKD3 (Ad-mycPKD3ca) using PKD-substrate motif LxRxx[S*/T*]-specific antibody (n = 3 independent experiments). **(C)** Scatter plot of the statistical significance of log2 transformed protein ratios versus log10-transformed label-free quantification intensities between control and PKD3ca expressing hepatocytes. Enriched proteins are indicated by red (2, significantly enriched) or green (1, potentially enriched) dots, blue ones are not enriched (0, not enriched). **(D)** 15 most enriched proteins identified by mass spectrometry showing a number, UniProt accession number, gene names, protein names, peptide count in EGFP and PKD3ca samples, respectively, and log2-transformed label-free quantification protein ratio are shown in the table. The detailed list of proteins identified are listed in Table S1.

showed that 11 sites of the motifs were previously reported. Noteworthy, three of the motifs were identified in the field of PKDs (also PKA signaling), namely, LsRklS16 for phenylalanine hydroxylase (PAH), and LtRqkS3894 as well as LtRqlS5407 for dystonin (DST) (Fig 3D). Moreover, a prediction tool for biological processes (ARCHS4) suggests that PKD3 signaling might influence the catabolic and metabolic activity of PAH (Fig 3E). PAH converts Phe to Tyr (Kaufman, 1958); therefore, these data suggest that PKD3 might also regulate AAs metabolism in the liver.

## PKD3 signaling determines tyrosine levels in the liver

To test whether PKD3 regulates PAH phosphorylation in hepatocytes, we transduced primary hepatocytes with increasing amounts of adenoviruses expressing either EGFP or PKD3ca. Subsequently, protein lysates were used to evaluate PAH migration on the SDS–PAGE followed by Western blotting. Interestingly, overexpression of PKDca leads to an upshift of PAH signal and appearance of the second band, which is specific for this protein. Interestingly, the upshift of PAH was more pronounced when hepatocytes were transduced with an increasing amount of PKD3ca (Fig 4A). Furthermore, to explore the physiological role of PKD3 in Phe metabolism, we cultivated hepatocytes expressing either EGFP or PKD3ca in the medium deprived of Phe and Tyr. Following Phe/Tyr starvation, we supplemented the cell culture medium of hepatocytes with increasing amounts of Phe and determined the tyrosine levels in the cells. In the cells, which were not supplemented with Phe, expression of PKD3ca resulted in the most pronounced increase in Tyr levels compared to control hepatocytes. Supplementation of Phe in the medium resulted in increased concentrations of Tyr levels in control hepatocytes. However, at each of the tested conditions, Tyr levels were significantly higher in the cells expressing PKD3ca but not increasing further upon the addition of Phe (Fig 4B). This suggests that PKD3 promotes the conversion of Phe to Tyr in hepatocytes. To test whether PKD3 regulates levels of Tyr in the complex in vivo situation, we measured Tyr levels in mice expressing PKD3ca specifically in hepatocytes (Mayer et al, 2019). Of note, mice overexpressing PKD3ca presented higher levels of Tyr in hepatic extracts than corresponding control animals (Fig 4C). Moreover, as revealed by metabolomics analysis, mice overexpressing PKD3ca

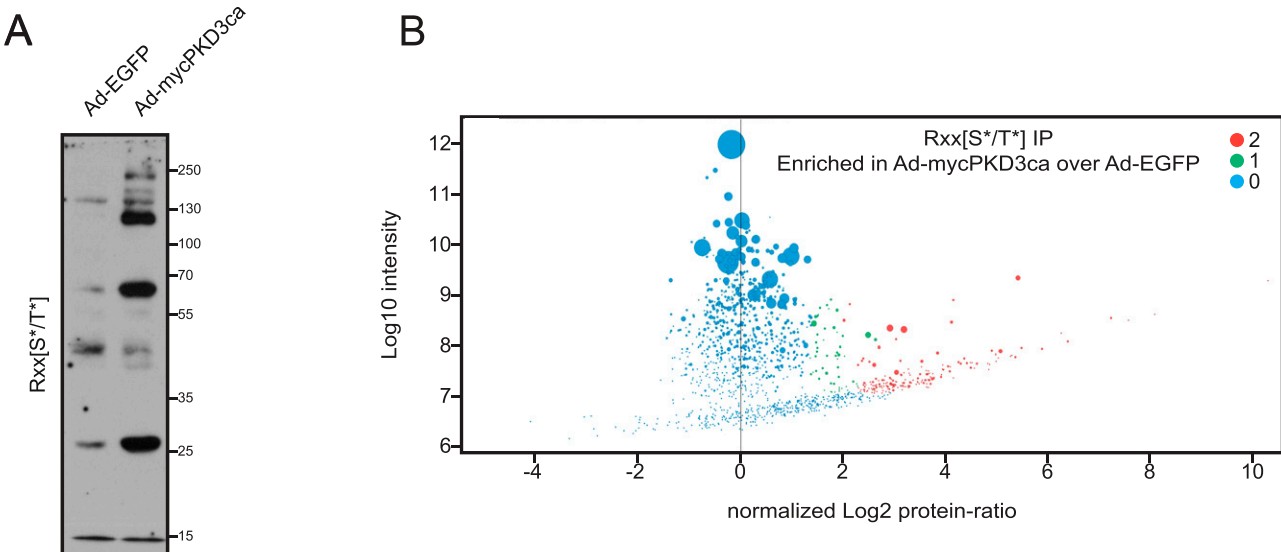

**Figure 2. Identification of protein kinase D (PKD)3 targets in hepatocytes by IP with Rxx[S*/T*] antibody.**
**(A)** WB using Rxx[S*/T*] motif antibody on lysates from PKD3-deficient hepatocytes expressing control EGFP (Ad-EGFP) or constitutive active form of PKD3 (Ad-mycPKD3ca).
**(B)** Scatter plot of the statistical significance of log2 transformed protein ratios versus log10-transformed label-free quantification intensities between control and PKD3ca expressing hepatocytes. Enriched proteins are indicated by red (2, significantly enriched) or green (1, potentially enriched) dots, blue ones are not enriched (0, not enriched).
**(C)** 15 most enriched proteins identified by mass spectrometry in the experiment from (C), showing a number, UniProt accession number, gene names, protein names, peptide count in EGFP and PKD3ca samples, respectively, and log2-transformed label-free quantification protein ratio. The detailed list of proteins identified are listed in Table S2.

presented also a higher Tyr to Phe ratio compared to control animals, while the levels of other AAs were not altered (Fig 4D and Supplemental Data 1). To test if PKD3 stimulates Tyr levels in the PAH-dependent manner, we have incubated PKD3ca expressing hepatocytes with Panobinostat a PAH inhibitor. Of note, inhibition of PAH decreased PKD3-induced Tyr levels (Fig 4E). Altogether, these results suggest that PKD3 has a key role in the conversion of Phe into Tyr in the liver.

### A glucagon-PKD3 axis determines amino acid metabolism in the liver

Seminal articles in the early 1970s demonstrated that hepatic PAH activity is hormonally regulated by glucagon via PKA signaling (Abita et al, 1976; Donlon & Kaufman, 1978). PKA phosphorylates Ser16 of

PAH upon glucagon stimulation, which increases the affinity of PAH for its main substrate, the amino acid Phe (Miranda et al, 2002). In addition, glucagon promotes the formation of DAG, a well-known activator of PKD3 (Hermsdorf et al, 1989; Rodgers, 2012). Glucagon is a classical hormone-induced under fasting conditions that rewires metabolism in the liver primarily via PKA (Pilkis et al, 1988; Pilkis & Granner, 1992). Thus, to investigate whether glucagon also governs PKD activity, we have injected fasted mice with glucagon. Of note, glucagon increased the abundance of active PKD and also PKD3 in the liver (Fig 4F). Importantly, PKD activity was almost completely abolished in PKD3-deficient mice (Fig 4F). These suggest that PKD3 might be required for glucagon-induced conversion of Phe to Tyr. Indeed, stimulation of primary control hepatocytes with glucagon resulted in increased Tyr levels, but in the cells-derived from

**A**

LxRxx[S*/T*]    Rxx[S*/T*]

60    24    202

**B**

LxRxx[S*/T*]    Rxx[S*/T*]

34 (55%)    12 (50%)    112 (55%)

Hits with putative PKD substrate motif [L/V/I]xRxx[S*/T*]

**C**

| No. | UniProt | Gene names | Protein names | LxRxx[S*/T*] Peptides Ad-EGFP | LxRxx[S*/T*] Peptides Ad-PKD3ca | LxRxx[S*/T*] Normalized Log2 protein-ratio | Rxx[S*/T*] Peptides Ad-EGFP | Rxx[S*/T*] Peptides Ad-PKD3ca | Rxx[S*/T*] Normalized Log2 protein-ratio | Putative PKD motif |
|---|---|---|---|---|---|---|---|---|---|---|
| 1 | Q5FWX6 | Pkd3 | Serine/threonine-protein kinase D3 | 0 | 46 | 9.55 | 3 | 42 | 5.38 | (1x) |
| 2 | Q9JKP5 | Mbnl1 | Muscleblind-like protein 1 | 1 | 2 | 7.73 | 1 | 2 | 7.53 | |
| 3 | F8VQB6 | Myo10 | Unconventional myosin-X | 2 | 14 | 4.13 | 0 | 24 | 3.72 | (1x) |
| 4 | Q8BHL4 | Gprc5a | Retionoic Acid-induced protein 3 | 0 | 2 | 4.03 | 0 | 3 | 4.46 | (1x) |
| 5 | P11352 | Gpx1 | Glutathione peroxidase 1 | 8 | 10 | 3.50 | 4 | 12 | 4.13 | |
| 6 | Q3UJB9 | Edc4 | Enhancer of mRNA-decapping protein 4 | 1 | 13 | 3.43 | 0 | 12 | 4.22 | |
| 7 | P70275 | Sema3e | Semaphorin-3E | 4 | 11 | 3.21 | 1 | 14 | 5.85 | |
| 8 | P13020 | Gsn | Gelsolin | 9 | 14 | 3.07 | 6 | 10 | 3.27 | |
| 9 | Q8BQ30 | Ppp1r18 | Phostensin | 5 | 15 | 2.91 | 4 | 23 | 4.09 | (1x) |
| 10 | P35564 | Canx | Calnexin | 2 | 4 | 2.49 | 1 | 7 | 3.16 | |
| 11 | Q61699 | Hsph1 | Heat shock protein 105 kDa | 1 | 6 | 2.49 | 0 | 7 | 3.35 | |
| 12 | Q91YU6 | Lzts2 | Leucine zipper putative tumor suppressor 2 | 1 | 4 | 2.36 | 0 | 8 | 3.64 | |
| 13 | Q62266 | Sprr1a | Cornifin-A | 14 | 10 | 2.07 | 2 | 9 | 5.45 | |
| 14 | P16331 | Pah | Phenylalanine-4-hydroxylase | 11 | 16 | 1.91 | 0 | 9 | 2.56 | (2x) |
| 15 | P61967 | Ap1s1 | AP-1 complex subunit sigma-1A | 1 | 2 | 1.97 | 0 | 3 | 2.31 | |
| 16 | P09055 | Itgb1 | Integrin beta-1 | 2 | 3 | 1.92 | 1 | 5 | 3.11 | |
| 17 | Q91VY9 | Znf622 | Zinc finger protein 622 | 2 | 2 | 1.91 | 0 | 8 | 3.43 | (2x) |
| 18 | P20918 | Plg | Plasminogen | 3 | 14 | 1.64 | 2 | 12 | 4.39 | (2x) |
| 19 | Q0VG62 | 1810022K09Rik | Uncharacterized protein C8orf59 homolog | 2 | 2 | 1.52 | 1 | 2 | 2.95 | (1x) |
| 20 | A0A087WQ89 | 2210011C24Rik | MCG5930 | 4 | 6 | 1.49 | 1 | 9 | 2.97 | (1x) |
| 21 | Q8K2I2 | Cchcr1 | Coiled-coil alpha-helical rod protein 1 | 0 | 4 | 1.44 | 0 | 5 | 2.62 | (2x) |
| 22 | P35700 | Prdx1 | Peroxiredoxin-1 | 7 | 10 | 1.41 | 3 | 11 | 7.19 | (1x) |
| 23 | Q91YD6 | Vill | Villin-like protein | 8 | 12 | 1.24 | 7 | 14 | 1.49 | |
| 24 | Q91ZU6 | Dst | Dystonin | 18 | 40 | 1.20 | 9 | 65 | 3.16 | (15x) |

**D**

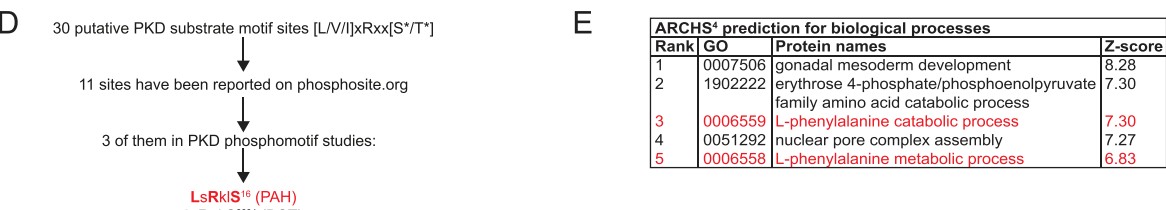

30 putative PKD substrate motif sites [L/V/I]xRxx[S*/T*]

↓

11 sites have been reported on phosphosite.org

↓

3 of them in PKD phosphomotif studies:

**L**sR**k**l**S**[16] (PAH)
**L**tR**q**k**S**[3894] (DST)
**L**tR**q**l**S**[5407] (DST)

**E**

| ARCHS[4] prediction for biological processes | | |
|---|---|---|
| **Rank** | **GO** | **Protein names** | **Z-score** |
| 1 | 0007506 | gonadal mesoderm development | 8.28 |
| 2 | 1902222 | erythrose 4-phosphate/phosphoenolpyruvate family amino acid catabolic process | 7.30 |
| 3 | 0006559 | L-phenylalanine catabolic process | 7.30 |
| 4 | 0051292 | nuclear pore complex assembly | 7.27 |
| 5 | 0006558 | L-phenylalanine metabolic process | 6.83 |

**Figure 3.  Comparative analysis of putative protein kinase D (PKD)3 targets identified using different motif antibodies.**
**(A)** Venn diagram of common putative substrates that were significantly enriched by both antibodies LxRxx[S*/T*] and Rxx[S*/T*]. **(B)** Computational identification of putative PKD motifs [L/V/I]xRxx[S*/T*] among putative substrates identified by both antibodies (percentage of proteins possessing a putative motif in brackets). Results were obtained using ExPASy ScanProsite tool. **(C)** List of 24 proteins enriched by both antibodies, LxRxx[S*/T*] and Rxx[S*/T*] with UniProt accession numbers, gene and protein names, peptide count, and log2 transformed label-free quantification protein ratio for LxRxx[S*/T*] and Rxx[S*/T*] immunoprecipitations, respectively, and the amount of putative PKD motifs for each protein. **(D)** Flow of computational identification of putative PKD substrates by comparing the PKD motifs within proteins using phosphosite.org repository. **(E)** Prediction of biological processes potentially regulated by PKD3 using ARCHS[4].

PKD3-deficient mice, glucagon failed to increase Tyr levels (Fig 4G). Next, we have tested if short term inhibition PKDs in isolated hepatocytes affects Tyr levels. In fact, PKD-specific inhibitor CTR0066101 ameliorated glucagon-induced Tyr levels in hepatocytes (Fig 4H). However, in mice pretreated for 5 d with CTR0066101 and challenged with glucagon, hepatic Tyr levels were not affected compared with the control animals (Fig 4I). All of these data suggest that activation of PKD3 by glucagon is required for induction of Phe to Tyr conversion in hepatocytes.

## PKD3 promotes PKA activity in the liver

PKA was previously shown to phosphorylate and activate PAH (Miranda et al, 2002). To test if PKD3 also directly phosphorylates PAH, we have performed an in vitro phosphorylation assay. As previously shown PKA phosphorylated recombinant PAH. However, PKD3 failed to phosphorylate PAH (Fig 5A), suggesting that PKD3 indirectly affects PAH function. Next, we tested if PKD3 affects PKA activity. The WB using an antibody directed against the phosphorylated motive of PKA (RRX*S/T) (Soundarapandian et al, 2020) revealed that PKA activity is increased in hepatocytes expressing PKD3ca (Fig 5B). Similarly, phosphorylation of PKA downstream target, CREB on Serine 133 (Rosenberg et al, 2002), was elevated in cells expressing PKD3ca (Fig 5B), which also indicates increased activity of PKA in hepatocytes. Of note, in livers of mice treated with PKD inhibitor (CTR0066101), at the dose which effectively inhibited PKD (Fig 5C), phosphorylation of PKA specific motif and catalytic subunit of PKA on serine 197 (Cauthron et al, 1998), were reduced (Fig 5C–E). Consistently, silencing of PKD3 in isolated primary hepatocytes, using specific shRNA (Fig 5F), was also

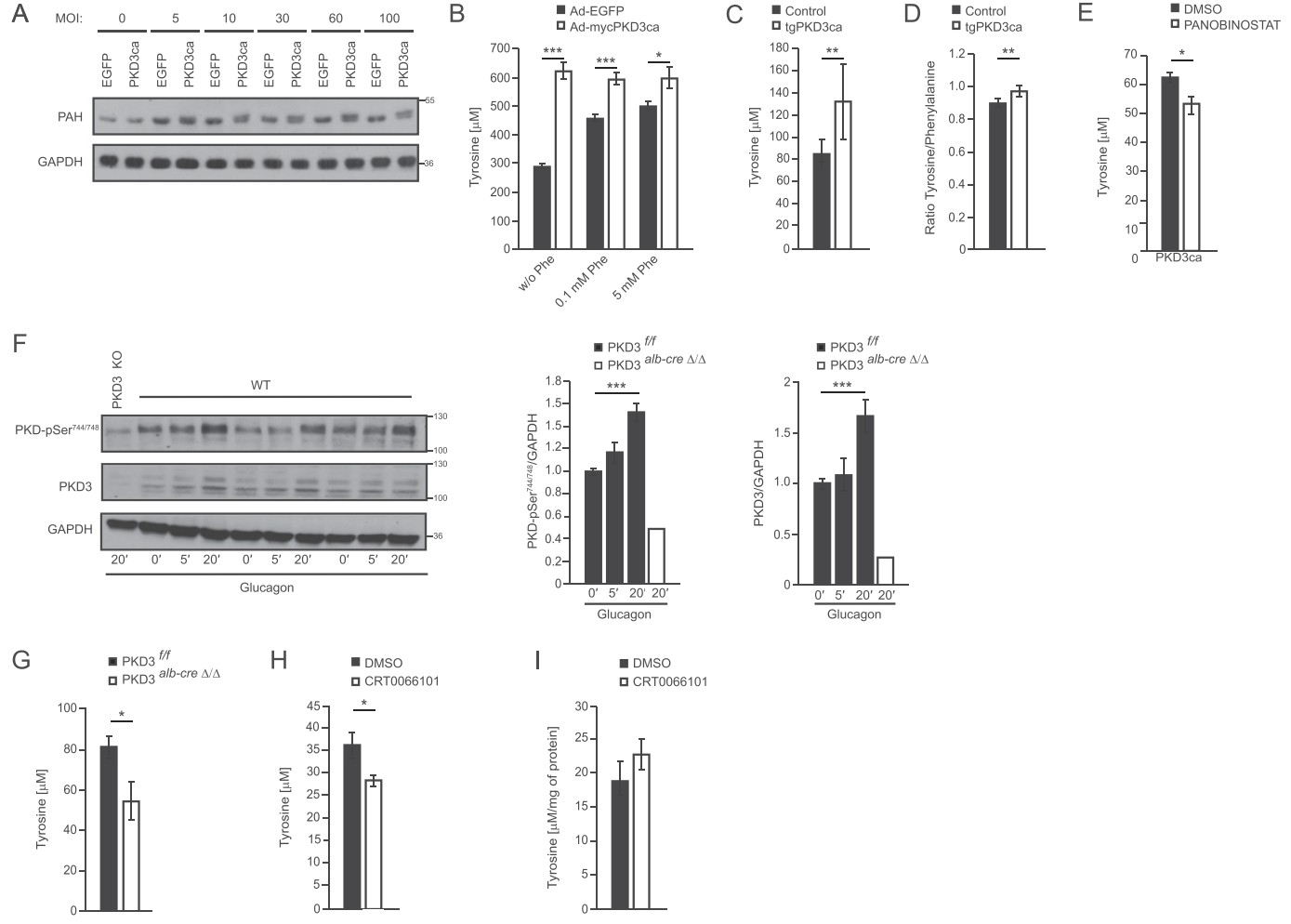

**Figure 4. Protein kinase D (PKD)3 signaling controls PAH activity and determines cellular levels of Tyr.**
**(A)** PAH expression and shifting analyzed in hepatocytes transduced with increasing amounts of adenovirus expressing EGFP or PKD3ca at indicated MOIs using WB. **(B)** Phe to Tyr conversion assay in primary hepatocytes expressing either EGFP or PKD3ca. Cells were depleted from Phe and Tyr in the medium for 1 h before stimulation and incubated with 0, 0.1, or 5 mM Phe for 1 h. (n = 4 per each group) **(C)** Tyr levels in livers from control mice and mice overexpressing PKD3ca (n = 12 and n = 16). **(D)** Tyr to Phe ratio in liver tissues from control mice and mice overexpressing PKD3ca (n = 12 and n = 16). **(E)** Intracellular Tyr levels in hepatocytes overexpressing PKD3ca and treated with Panobinostat 2 μM for 24 h (a PAH inhibitor) as indicated (n = 4) **(F)** WB analysis of liver extracts isolated from mice fasted for 6 h and stimulated with glucagon at the dose of 200 μg/kg of body weight (i.p. injection) for indicated time points using antibodies against p-PKD, PKD3 (upper band corresponds to the predicted, molecular weight of PKD3), and GAPDH. **(G)** Tyr levels in primary hepatocytes isolated from control and PKD3-deficient mice and stimulated with glucagon (200 nM) for 20 min. **(H)** Tyr levels in primary hepatocytes treated with CRT0066101 (1 μM) 2 h prior stimulation with glucagon (200 nM) for 20 min. **(I)** Tyr levels in the livers of mice treated with CRT0066101 inhibitor for 5 d daily (10 mg/kg of body weight, i.p injection), fasted for 6 h, and challenged with glucagon for 10 min at the dose of 200 μg/kg of body weight (n = 6 per group). Data are presented as mean ± SEM. *P > 0.05, ***P > 0.001 (t test and one-way ANOVA with post hoc Tukey's test). Source data are available for this figure.

sufficient decrease phosphorylation of PKA specific motive (Fig 5G). Of note, PKD inhibitor did not significantly decrease PKA activity in the absence of PKD3 (Fig 5G). All of these indicate that PKD3 promotes PKA activity in hepatocytes.

PKA is a master regulator of the hepatic response to the fasting-induced hormone glucagon. PKA stimulates the process of glyco-genolysis and gluconeogenesis and therefore maintains glucose levels during food deprivation (London et al, 2020). To test if PKD3 contributes to the response to glucagon and fasting, we used mice treated with CTR0066101 inhibitor as well as animals deficient for PKD3 in hepa-tocytes. Importantly, mice treated with CTR0066101 for 5 d and fasted for short time (6 h) presented markedly lower glucose levels when

challenged with glucagon (Fig 5H). Consistently, mice deficient for PKD3 in hepatocytes presented lower glucose levels when fasted for a prolonged period (Fig 5H). All of these indicate that PKD3 regulates PKA activity to promote a broad spectrum of hepatic metabolism.

# Discussion

Recent studies established PKD3 function in the regulation of hepatic lipid and glucose metabolism (Mayer et al, 2019). However, the phosphorylation targets of PKD3 in hepatic cells remained elusive. Utilizing a proteomic approach on primary hepatocytes deficient for

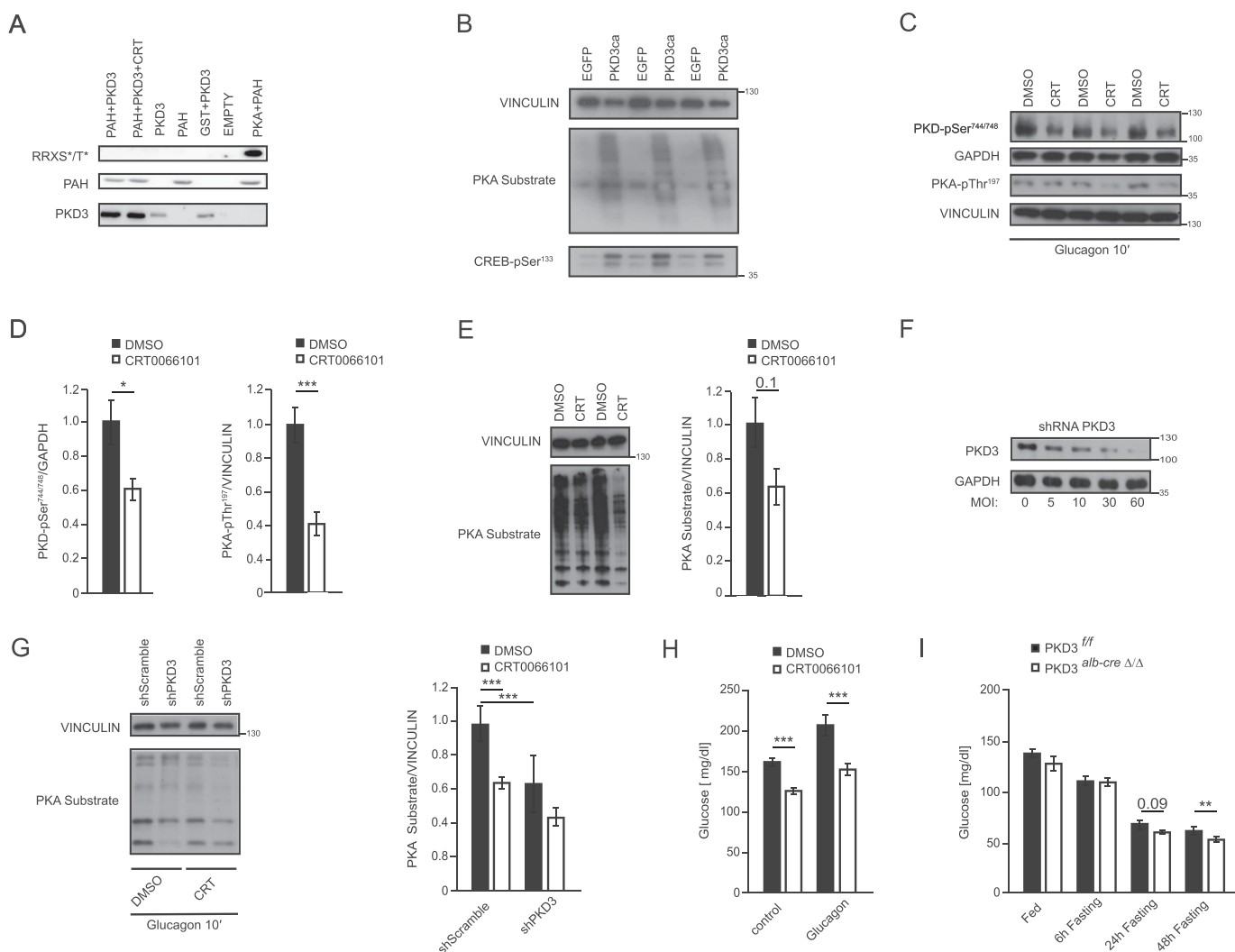

**Figure 5. Protein kinase D (PKD)3 promotes fasting response by targeting PKA activity.**
**(A)** In vitro kinase assay using recombinant, PAH, PKA, and PKD3 as indicated on the figure the phosphorylation on PAH was assessed using RXX*S/T antibody. **(B)** WB for indicated proteins in primary hepatocytes overexpressing PKD3ca and control cells (n = 3). **(C, D, E)** WB for indicated proteins and corresponding quantifications on extracts isolated from liver of mice treated with CRT0066101 inhibitor for 5 d (10 mg per kg of body weight, i.p. injection) and corresponding control animals fasted for 6 h and euthanized 10 min after glucagon injection (200 µg/kg of body weight) (n = 3). **(F)** WB for PKD3 and loading control GAPDH on extracts isolated from primary hepatocytes transduced with increasing MOI of adenoviral particles containing shRNA targeted against PKD3. **(G)** WB for indicated proteins and corresponding quantification of WBs on protein extracts isolated from primary hepatocytes transduced with adenoviral particles containing shRNA against PKD3 or control shRNA and treated with CRT0066101 inhibitor or DMSO as vehicle. **(H)** Blood glucose levels before and after 10 min of glucagon injection at the dose of 200 µg/kg of body weight in mice treated with CRT0066101 inhibitor for 5 d (10 mg/kg of body weight). **(I)** Mice were fasted for 6 h before glucagon (I) Blood glucose levels in mice deficient for PKD3 in hepatocytes and corresponding control animals fasted for the indicated times (n = 6 per group). Data are presented as mean ± SEM. *$P > 0.05$, ***$P > 0.001$ (one-way ANOVA with post hoc Tukey's test or $t$ test for comparison of two groups).
Source data are available for this figure.

PKD3 and re-expressing PKD3ca we identified more than 300 putative direct or indirect targets of PKD3. Importantly, this approach resulted in the identification of the novel function of PKD3 in the regulation of hepatic metabolism of AAs. Namely, we showed that PKD3 promotes the conversion of Phe to Tyr by activating PAH, a rate-limiting enzyme in this process. Moreover, we have linked PKD3 action in the liver to the induction PKA signaling which regulates hepatic response to glucagon and fasting.

In our studies, we used two complementary proteomic approaches. For pull-downs, we used two antibodies: against the full phospho-motif sequence of PKD LxRxx[S*/T*] and antibody against part of the PKD phospho-motif sequence Rxx[S*/T*]. Importantly, pull-downs using an antibody against part of the PKD motif revealed more of the potential targets of PKD3 than an antibody against the full sequence targeted by PKDs. This indicates that in a large number of PKD3 target proteins AA at the position −5 in relation to the phospho-acceptor AA might be other than leucine. Whether we used for pull-downs antibody against LxRxx[S*/T*] or Rxx[S*/T*] motif, almost 50% of identified proteins did not present in their sequence the consensus motif for PKDs. These might

indicate that in our pull-downs we have also fished out proteins, which are interacting with the targets of PKD3 but are not phosphorylated by PKD3 themselves. Moreover, using this approach, we cannot exclude that changes in the phosphoproteome evoked by manipulations of PKD3 activity are the consequence of activation/suppression of other kinase/s, which might target the sequence of AAs recognized by both antibodies used in this study.

In total, 12 proteins were identified by both pull-downs and possess one or more PKD phosphorylation motifs in their sequence. Among them, we found PAH as a target for PKD3, which we confirmed by classical Western blot (WB). In line with our findings, a computational analysis to predict gene functions suggests that PKD3 might be involved in phenylalanine metabolism. PAH has two putative PKD motifs (LsRklS16 and IpRpfS411). It was shown that Ser16 is phosphorylated upon glucagon stimulation via PKA activation (Abita et al, 1976; Donlon & Kaufman, 1978). Furthermore, the classical PKA motif is RRx[S*/T*] and has in −2 position of Ser16 a lysine (K) and not arginine (R), which can also serve as a PKD motif (Pinna & Ruzzene, 1996). Nevertheless, in vitro kinase assay revealed that PKD3 does not phosphorylate PAH directly like PKA. Further analyzes revealed that PKD3 stimulates PKA activity in hepatocytes. However, the mechanisms responsible for PKD3 induced PKA activity remain elusive. Therefore, it is plausible that glucagon might affect hepatic PAH activity via PKD3-PKA signaling leading to changes in Tyr concentration. Phosphorylation of PAH on Ser16 increases the affinity of this enzyme to Phe (Døskeland et al, 1996; Arun et al, 2011). Consistently, primary hepatocytes overexpressing PKD3ca presented higher levels of Tyr especially when cells were starved from Phe. Importantly, transgenic mice overexpressing PKD3ca had higher levels of Tyr in the liver as well as a higher Tyr to Phe ratio than littermate controls.

Our findings suggest that glucagon also promotes the activation of PKD3. Moreover, activation of PKD3 is required for the induction of conversion of Phe to Tyr by glucagon in isolated hepatocytes. However, inhibition of PKD activity in mice using CTR0066101 compound did not affect hepatic Tyr levels. This might indicate that in vivo other factors than the rate of Phe to Tyr conversion could affect Tyr levels in the liver. Interestingly, in line with the PKA action in hepatocytes, we have found that PKD3 promotes fasting and glucagon-induced glucose levels in mice. This indicates that PKD3 evoked PKA activity regulates a broader spectrum of hepatic metabolism. All of these findings suggest that glucagon might act in the liver also in a PKD3-dependent manner.

As mentioned above, PKD3 regulates insulin signaling and lipogenesis in the liver by modulation of mTORC1, mTORC2, and AKT signaling (Mayer et al, 2019). Our current proteomic approach identified several potential targets of PKD3, which could explain the suppression of mTORC1, mTORC2, and AKT signaling by PKD3. For instance, Ral GTPase activating protein non-catalytic $\beta$ subunit (RalGAP$\beta$), which can control mTORC1 activity in response to insulin stimulation (Chen et al, 2011; Martin et al, 2014), was the most enriched protein in the Rxx[S*/T*]-motif antibody-based pull-down. Moreover, Tuberous sclerosis (TSC) 1 and 2, which also control mTORC1 and mTORC2 activity (Ben-Sahra & Manning, 2017), were also found using our strategy to be a putative target of PKD3. Interestingly, the distal downstream target of mTORC2, NDRG1, was also found to be a putative target of PKD3. Because previous studies showed that NDRG1

promotes lipogenesis (Cai et al, 2017), this might also explain PKD3-dependent lipogenesis in the liver. However, these putative targets of PKD3 require confirmation and the detailed functions of PKD3-dependent phosphorylation needs to be identified.

In different cancer cell types, GIT1, S6K1, and PFKFB3 have been identified as targets of PKD3 (Huck et al, 2012, 2014; Laplante & Sabatini, 2012; Zhang et al, 2019). However, these proteins did not appear in our pull-downs. This might indicate that in hepatocytes PKD3 phosphorylates a different set of proteins in respective cancer cell types.

In summary, in our study, we identified a plethora of putative targets for PKD3 in the liver. Among them PAH, which suggests that PKD3 plays a role in AAs metabolism. We confirmed that PKD3 promotes the conversion of Phe to Tyr in response to glucagon stimulation. Our data also indicate that PKD3 stimulates fasting glucose levels in mice. Moreover, we have identified numerous putative targets, which might suggest the role of PKD3 in the regulation of lipid metabolism or in insulin-dependent signaling.

# Materials and Methods

### Primary hepatocyte isolation, culture, and infection

Primary mouse hepatocytes were prepared as described previously (Mayer et al, 2019). All relevant mouse models of PKD3-deficiency or overexpression were also described in Mayer et al (2019). All animal studies were approved by the local animal welfare authorities (Regierung von Unterfranken) with the animal protocol no. AK55.2-2531.01-124/13 and 55.2-2532-2-741. All mouse primary hepatocytes were infected 4–6 h after plating with adenoviruses expressing either EGFP (Ad-EGFP) or a constitutively active form of PKD3 (Ad-mycPKD3ca) shRNA PKD3 (ad-shRNA-PKD3) at a MOI of 10, 30, and 60, respectively. Adenoviruses were purchased from Vector Builder. Medium was replaced the following morning, and cells were used for experiments 36–48 h after infection. Transduction efficiency, which was 100%, was assessed by analyzing the expression of the EGFP reporter (which was present in all adenoviruses).

### Phenylalanine conversion assay

Primary hepatocytes were fasted in DMEM without phenylalanine (Phe) and tyrosine (Tyr) for 1 h. Afterward, the cells were stimulated with either 0, 0.1, or 5 mM Phe for 1 h (500 $\mu$l/well, 12-well plate). Next, the cells were lysed in 120 $\mu$l lysis buffer followed by centrifugation at 10,000$g$ for 10 min at 4°C. Tyrosine was quantified using the Tyrosine Assay Kit (ABNOVA) according to the manufacturer's protocol.

### Amino acids and glucagon stimulation

Primary hepatocytes were fasted in DMEM without AAs for 1 h. Then they were cultured with either no amino acids, with all amino acids, all amino acids except Phe and Tyr, with Phe exclusively, or Phe and Tyr exclusively for 1 h. DMEM without AA and DMEM without Phe and Tyr were supplemented with respective amounts of glucose, serine,

glycine, Phe, and Tyr. Hepatocytes were stimulated with glucagon for 0, 5, and 20 min.

### Immunoprecipitation (IP) and WB analysis

IP was performed on hepatocyte (which were isolated from three animals and transduced as indicated in the specific figures) lysates using antibodies against LxRxx[S*/T*] and Rxx[S*/T*] phospho-motifs (both Cell Signaling Technology) with Pierce Protein A/G Magnetic Beads according to the manufacturer's protocol. Briefly, 4 mg of protein (2 mg/ml) and 30 µg of antibody were used for each IP. Samples were eluted in 1× NuPAGE lithium dodecyl sulfate sample buffer supplemented with 60 mM dichlorodiphenyltrichloroethane (DDT) for 10 min at 95°C. Beads were magnetically separated from the immunoprecipitated product, which was further analyzed on WB or by Mass spectrometry. Tissues samples were lysed in radioimmunoprecipitation assay buffer (RIPA buffer) and blotted using classical WB techniques. Antibodies against p-PKD3 S744/748, PKD3/PKCv, pPKA Thr197, pCREB Ser133, Vinculin, and PKA Substrate Antibody were purchased from Cell Signaling Technology. GAPDH antibody was purchased from Thermo.

### Mass spectrometry analysis

Gel electrophoresis and in-gel digestion were carried out according to the standard procedures.

An Orbitrap Fusion equipped with a PicoView ion source and coupled to an EASY-nLC 1,000 were used for Nano-LC-MS/MS analyzes.

MS and MS/MS scans were both obtained using an Orbitrap analyzer. The raw data were processed, analyzed, and quantified using the MaxQuant software. Label-free quantification (LFQ) intensities were used for protein quantification. Proteins with less than two identified razor and unique peptides were excluded. Data imputation was performed with values from a standard normal distribution with a mean of the 5% quantile of the combined log10-transformed LFQ intensities and a standard deviation of 0.1. Log2 transformed protein ratios of the sample versus control with values outside a 1.5× (potential, significance 1) or 3× (extreme, significance 2) *interquartile range* (IQR), respectively, were considered as significantly enriched.

### Metabolomics analysis (HPLC)

For AAs analysis, pieces of mouse liver were homogenized in 69 vol methanol/$H_2O$ (80/20, vol/vol) containing 3.5 µM lamivudine as external standard using an ultraturrax. The resulting homogenate was centrifuged (two min max rpm) and 600 µl supernatant was applied to an activated and equilibrated RP18 solid phase extraction column (activation with 1 ml acetonitrile and equilibration with 1 ml methanol/$H_2O$ [80/20, vol/vol] [Phenomenex Strata C18-E, 55 µm, 50 mg/1 ml; Phenomenex]). The resulting eluate was evaporated at room temperature using a vacuum concentrator. The evaporated samples were re-dissolved in 100 µl of 5 mM $NH_4OAc$ in acetonitrile/$H_2O$ (25/75, vol/vol). The equipment used for LC/MS analysis was a Thermo Fisher Scientific Dionex Ultimate 3000 UHPLC system hyphenated with a Q Exactive mass spectrometer equipped with a heated electrospray ionization (HESI) probe (Thermo Fisher Scientific). LC parameters were as follows: mobile phase A consisted of 5 mM $NH_4OAc$ in acetonitrile/$H_2O$ (5/95, vol/vol) and mobile phase B consisted of 5 mM NH4OAc in acetonitrile/$H_2O$ (95/5, vol/vol). Chromatographic separation of AAs was achieved by applying 3 µl of dissolved sample on a SeQuant ZIC-HILIC column (3.5 µm particles, 100 × 2.1 mm) (Merck), combined with a Javelin particle filter (Thermo Fisher Scientific), and a SeQuant ZIC-HILIC precolumn (5 µm particles, 20 × 2 mm) (Merck) using a linear gradient of mobile phase A (5 mM $NH_4OAc$ in acetonitrile/$H_2O$ (5/95, vol/vol)) and mobile phase B (5 mM NH4OAc in acetonitrile/$H_2O$ (95/5, vol/vol)). The LC gradient program was 100% solvent B for 2 min, followed by a linear decrease to 40% solvent B within 16 min, then maintaining 40% B for 6 min, then returning to 100% B in 1 and 5 min 100% solvent B for column equilibration before each injection. The column temperature was set to 30°C and the flow rate was maintained at 200 µl/min. The eluent was directed to the ESI source of the Q Exactive mass spectrometer from 1.85–20.0 min after sample injection. MS scan parameters were as follows: scan type: full MS, scan range: 69.0–1,000 m/z, resolution: 70,000, polarity: positive and negative, alternating, AGC-target: $3 × 10^6$, maximum injection time: 200 ms HESI. Source parameters were as follows: sheath gas: 30, auxiliary gas: 10, sweep gas: 3, spray voltage: 2.5 kV in pos. mode and 3.6 kV in neg. mode, capillary temperature: 320°C, S-lens RF level: 55.0, Aux gas heater temperature: 120°C. For data evaluation: peaks corresponding to the calculated amino acid masses (MIM ± H+ ± 2 mMU) were integrated using TraceFinder software (Thermo Fisher Scientific). Alternatively (for Figs 4B and C and 5B), commercially available kit was used for Tyr quantification (Cell Biolabs, Biocat).

### In vitro kinase assay

Recombinant GST-PAH (vector synthesized by Vectorbuilder) was produced in *Escherichia coli* (BL21) as GST fusion protein, and purified by affinity chromatography on glutathione–Sepharose columns. Recombinant human proteins, PKD3 and PKA, were both purchased from Enzo biosciences and SignalChem Biotech, respectively. Kinase reactions were performed in reaction buffer (Cell Signaling Technology) in the presence of cold (nonradioactive) ATP (Cell Signaling Technology) for 30 min at 30°C. As indicated in the experiment, 1 µM of CRT0066101 (Tocris) was added to the corresponding condition. Proteins from the kinase reactions were boiled in 5× Laemmli buffer and analyzed by Western blotting. Membrane was incubated with the appropriate primary antibody against the phosphorylated motif (RxxS/T*) (Cell Signaling Technology), PAH (proteintech, PK), and PKD3 (Cell signaling Technology).

## Data Availability

The mass spectrometry proteomics data haven deposited to the ProteomeXchange Consortium via the PRIDE partner repository with the data set identifier PXD026599.

## Supplementary Information

# Acknowledgements

We thank Dr Olga Sumara for critical reading of our manuscript. This study was funded by European Research Council Starting Grant SicMetabol (no. 678119), Emmy Noether grant (Su 820/1-1) from the German Research Foundation, European Molecular Biology Organization (EMBO) Installation Grant from EMBO, and the Dioscuri Centre of Scientific Excellence. The program was initiated by the Max Planck Society, managed jointly with the National Science Centre and mutually funded by the Ministry of Science and Higher Education (MNiSW) and the German Federal Ministry of Education and Research (BMBF).

## Author Contributions

A Loza-Valdes: conceptualization, data curation, formal analysis, investigation, methodology, and writing—original draft, review, and editing.
AE Mayer: conceptualization, data curation, formal analysis, investigation, methodology, and writing—original draft.
T Kassouf: conceptualization, data curation, and investigation.
J Trujillo Viera: conceptualization, data curation, and investigation.
W Schmitz: conceptualization, data curation, and formal analysis.
F Dziaczkowski: investigation and methodology.
M Leitges: resources.
A Schlosser: conceptualization, resources, data curation, and formal analysis.
G Sumara: conceptualization, resources, data curation, formal analysis, funding acquisition, investigation, methodology, project administration, and writing—original draft, review, and editing.

## Conflict of Interest Statement

The authors declare that they have no conflict of interest.

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
