## [Reviewer comments · Life Science Alliance]

Life Science Alliance

A phosphoproteomic approach reveals that PKD3 controls PKA-mediated glucose and tyrosine metabolism

Angel Loza Valdes, Alexander Mayer, Toufic Kassouf, Jonathan Trujillo Viera, Werner Schmitz, Filip Dziaczkowski, Michael Leitges, Andreas Schlosser, and Grzegorz Sumara

DOI: <https://doi.org/10.26508/lsa.202000863>

Corresponding author(s): Grzegorz Sumara, Rudolf Virchow Center for Experimental Biomedicine University of Würzburg

Review Timeline:

Submission Date:	2020-07-28
Editorial Decision:	2020-09-11
Revision Received:	2021-02-24
Editorial Decision:	2021-04-07
Revision Received:	2021-05-11
Editorial Decision:	2021-05-19
Revision Received:	2021-06-09
Accepted:	2021-06-10

Transaction Report:

September 11, 2020

Re: Life Science Alliance manuscript #LSA-2020-00863-T

Dr. Grzegorz Sumara
Rudolf Virchow Center for Experimental Biomedicine University of Würzburg
Josef-Schneider-Straße 2, Haus D15
Würzburg 97080
GERMANY

Dear Dr. Sumara,

Thank you for submitting your manuscript entitled "A phosphoproteomic approach reveals that PKD3 controls phenylalanine and tyrosine metabolism" to Life Science Alliance (LSA). We apologize for this delay in getting back to you.

The manuscript has been reviewed by the editors and outside referees (reviewer comments below). As you will see, the reviewers were quite enthusiastic about the study and its potential impact, but have raised several concerns that should be addressed prior to further consideration of the manuscript at LSA. Therefore, we would encourage you to submit a revised version that addresses the referees' concerns, particularly the ones pertaining to the glucagon-PKD3-PAH axis section of the manuscript.

We would be happy to discuss the individual revision points further with you should this be helpful. While you are revising your manuscript, please also attend to the below editorial points to help expedite the publication of your manuscript. The typical timeframe for revisions is three months. Please note that papers are generally considered through only one revision cycle, so strong support from the referees on the revised version is needed for acceptance. When submitting the revision, please include a letter addressing the reviewers' comments point by point.

Thank you for this interesting contribution to Life Science Alliance. We are looking forward to receiving your revised manuscript. Please direct any editorial questions to the journal office.

Sincerely,

Shachi Bhatt, Ph.D.
Executive Editor
Life Science Alliance

B. MANUSCRIPT ORGANIZATION AND FORMATTING:

Reviewer #1 (Comments to the Authors (Required)):

In this paper, the authors aim to identify downstream targets and signaling pathways of the serine/threonine kinase PKD3 in primary hepatocytes in vitro and in vivo. The PKD family comprises three isoforms (PKD1-3), the functions of which depend on the cell type and external signal cues. While the functions of the kinases has been extensively studied in vitro, knowledge of their physiological role is still limited. This paper thus addresses an important issue in the field.

The authors employ two different phosphoproteomic approaches in which they compare primary PKD3 co-hepatocytes transduced with either adenovirus encoding EGFP or myc-labeled active PKD3. Using two different antibodies specific for phosphorylated PKD substrate motifs, several proteins were identified that showed increased phosphorylation at potential PKD phosphorylation sites in caPKD3-expressing hepatocytes compared to control cells. Among these, phenylalanine hydroxylase (PAH) appeared to be the most promising target. In agreement with this, the authors show that PKD3 is involved in the conversion of phenylalanine to tyrosine in vitro and in vivo.

The paper sheds light on a possible new function of PKD3 in liver metabolism and adds important information on isoform-specific functions of this member of the PKD family. While the data showing the involvement of PKD3 in tyrosine and phenylalanine metabolism in vitro and in vivo are strong and convincing, the experiments on PKD3-dependent phosphorylation of PAH and its functional

connection with the metabolic phenotype of PKD3 co-mice are inadequate.

Major points need to be addressed here before the paper can be considered for publication.

Figure 3: As the authors point out, PKA has been identified as an upstream kinase of serine 16 in PAH in previous publications and phosphorylation of this site increases the affinity of PAH for its substrate Phe. Consequently, as serine 16-phosphorylated PAH was pulled down with a PKD pMOTIF substrate antibody the authors conclude that PAH could be a substrate for PKD3 as well. Unfortunately, this is not being pursued consistently by the authors.

Thus, in order to link PAH activity to PKD3 kinase activity and to exclude or prove that serine 16 in PAH is a direct PKD3 target, an in vitro kinase assay must be performed. If proven that PKD3 phosphorylates PAH directly, some discussion on whether the two kinases, PKD3 and PKA, compete for PAH phosphorylation and how this might be regulated would be expected. Also, in case PAH is not a direct target of PKD3 it would be interesting to have some thoughts about how PKA and PKD3 signaling might be linked.

Figure 4: The authors propose a PKD3-dependent regulation of serine 16 phosphorylation in PAH. This assumption is based on a mobility shift of PAH upon PKD3ca expression. However, whether this mobility shift is related to serine 16 phosphorylation remains unclear. The PKD-pMOTIF substrate antibody that pulled down PAH should be used to demonstrate enhanced phosphorylation of endogenous PAH upon PKD3 expression in PKD3 co-cells in a Western blot analysis. This should be accompanied by an experiment that proves that this antibody detects the phosphorylation of PAH on serine 16. These experiments would also help to functionally link the phosphorylation and activity of PAH to PKD3.

Figure 5: The authors state that glucagon promotes PKD3 activation regardless of the AA composition. However, this is difficult to understand from the Western blot shown. The PKD3-phospho-Blot shows two bands at the expected size, but which of them corresponds to the active PKD3? To clarify, the authors should compare with lysates obtained from PKD3-Ko-cells. Furthermore, if it is the lower band, there are significant differences in the amount of detectable phosphorylated PKD3, which seem to be dependent on the AA composition. In particular, the exclusive Phe / Tyr condition seems to have a negative influence on the glucagon-induced PKD3 activation. A quantitative analysis of three independent experiments is required to better evaluate the data. In addition, the total PKD3 values must be displayed.

In addition, it would be important to show whether glucagon can promote the phosphorylation of PAH and the dephosphorylation of S6K and 4E-BP1 in PKD3 ko-hepatocytes. This would complement the data showing that the glucagon-induced increase in tyrosine levels is dependent on PKD3.

Minor points

Molecular weight markers on the blot in figure 4A are missing.

Reviewer #2 (Comments to the Authors (Required)):

The manuscript by Mayer et al. sheds new light on the action of PKD3 in the liver. The authors identify several putative targets of this kinase, and have further investigated on PAH and amino

acid metabolism in response to glucagon stimulation.

Supporting data are: the PAH phospho-protein was found in 2 independent phosphoproteomics experiments; PKD motifs were previously identified in the PAH sequence; tyrosine levels are increased in PKD3 overexpressing mouse hepatocytes in vivo and in primary cultures; glucagon promotes activation of PKD3; primary hepatocytes lacking PKD3 fail to accumulate tyrosine after glucagon stimulation.

Overall, the paper is well written and the direct link between PKD3 and amino acid metabolism is supported by data.

However, more evidence to support the glucagon-PKD3-PAH axis would be needed before publication:

- In Fig. 4B and C, the authors show tyrosine accumulation in hepatocytes and whole liver in PKD3 overexpression systems. Would that be still observed upon concomitant silencing of PAH?
- Would a PKD3 inhibitor affect the glucagon induced upshift of PAH and tyrosine levels (related to Fig.5A)?
- Primary PKD3-KO hepatocytes fail to increase tyrosine levels upon glucagon stimulation (Fig.5B); is this phenotype also seen in vivo, after glucagon stimulation or in fasting conditions?

Minor comment:

- Please provide the metabolomics data. Supplementary Table 2 refers to phosphoproteomics and Supplementary Table 3 is not included, actually.

Reviewer #3 (Comments to the Authors (Required)):

Mayer et al have applied proteomic approaches to determine PKD3 targets in hepatocytes using a phospho-motif antibody pull-down approach coupled with mass spectrometry.

The authors used adenoviral vectors to express either GFP (control) or PKD3ca (a constitutively active PKD3 mutant) in PKD3 KO hepatocytes before immunoprecipitating cellular proteins that are reactive with either LxRxx(pS/pT) or with Rxx(pS/pT) motif antibodies. Subsequent mass spec analysis identified 310 proteins that were enriched in the immunoprecipitates, and approximately half of these contained at least one putative PKD substrate motif, [L/V/I]xRxx[pS/pT]. Of these, 24 proteins were identified in both screens and were thus identified by the authors as putative PKD3 targets in murine hepatocytes.

The authors next focused on one putative PKD3 target that was identified by their IP/mass spec approach, phenylalanine hydroxylase (PAH), concluding that PKD3 is involved in controlling phenylalanine conversion to tyrosine in primary hepatocytes. PAH contains two putative PKD phosphorylation sites, one of which surrounds Ser16, a known regulatory site that influences the affinity of PAH for phenylalanine and which has previously been shown to be phosphorylated in response to glucagon in a PKA-dependent manner. In this manuscript the authors show that

expression of PKD3ca in PKD3 KO cells was sufficient to induce a MW shift in the PAH protein, indicative of a post-translational modification event. The authors next demonstrated, using an in vitro conversion assay, that PKD3ca expression boosted cellular tyrosine levels in PKD3 KO cells (compared to control GFP+ cells) under Tyr starvation conditions, independent of whether extracellular Phe was present or not. A similar result was observed for basal levels of tyrosine in the livers of PKD3ca transgenic mice versus control mice.

Finally, the authors investigated whether PKD3 regulates glucagon-induced phenylalanine metabolism, showing that glucagon treatment stimulates the conversion of Phe to Tyr in wild-type hepatocytes but not in PKD3 KO hepatocytes under conditions of Phe/Tyr starvation. This correlates with findings showing that a large MW shift in PAH is induced by glucagon in Phe/Tyr-starved cells (and that PKD3 T-loop phosphorylation is also increased by glucagon, albeit in an AA-independent manner).

This is a very nice study that advances the field in terms of how PKD3 is involved in glucagon signaling/cellular metabolism, as well as providing a great resource for further investigations. If the authors can address the issues highlighted below (by either including additional supporting data or by modifying their results analysis/discussion) then I recommend publication.

1.Expression of a constitutively active kinase can cause spurious results, due to dysregulated activity (spatial and/or temporal) - i.e. just because a constitutively active mutant can do something, does it really regulate that event in vivo? The authors do include key data in Figure 5 showing that deletion of PKD3 negatively impacts on Phe to Tyr conversion in cultured hepatocytes compared to wild-type cells. This data might be better placed earlier in the manuscript however to really highlight the essential role of PKD3 in this metabolic process.

2.Inclusion of additional data would strengthen the main message of this manuscript. e.g Is the PAH MW shift reduced in PKD3 KO cells vs wild-type cells? What effect do PKD specific inhibitors (and/or expression of a catalytically dead PKD3 mutant) have on glucagon-induced PAH MW shifts? Are tyrosine levels reduced in the livers of PKD3 KO vs wild-type mice under different feeding conditions?

3.The authors conclusion that glucagon regulates PKD3 T loop phosphorylation (Figure 5A) is not convincing as the identity of PKD3 amongst the protein species present is not clear (several do not appear to be glucagon regulated from the data shown) - cells from PKD3 KO mice could be used as a control perhaps?

4.How reproducible are the three biological replicates for the IP/mass spec experiments shown in Figures 1 and 2? This information should be provided.

5.The authors should introduce a note of caution when interpreting/discussing the results shown in Figures 1-3, with regard to the potential for non-specific interactions with the two phospho-motif antibodies used. e.g. Protein kinases other than PKDs can phosphorylate LxRxx(pS/pT) or Rxx(pS/pT) motifs, also some of these interactions may be occurring in a non-phospho-specific manner and thus could be influenced by changes in protein expression level.

6.As shown in Figure 4B, tyrosine levels increase in control (GFP+) PKD3 KO hepatocytes when extracellular Phe levels are increased, suggesting that PKD3 may not be the sole regulator of PAH and/or Phe to Tyr conversion?

7. The number of replicate experiments and specific details on the nature and results of any statistical tests, including specific p values, should be indicated for all experiments in the manuscript.

Response letter LSA-2020-00863-T

Dear Dr. Bhatt,

Thank you very much for your efforts to improve our manuscript.

We found the comments of all three reviewers very useful. Despite some difficulties in obtaining some of the reagents (for example PAH antibody) due to the current pandemic situation, in our opinion, we have managed to address all of the concerns which were raised. In fact, the experiments suggested by reviewers draw our attention to other aspects of PKD3 action in hepatocytes. Especially, comments of reviewer #1 allowed us to discover that PKD3 regulates PKA activity, the major signaling pathway activated by glucagon. These allowed us to expand our study from PAH-mediated tyrosine metabolism also to the regulation of glucose homeostasis by PKD3. Therefore, we made some substantial changes to our manuscript.

Below, you can find a point-by-point response to all the concerns raised by the reviewers.

Thank you once again for your help.

Best regards,

Grzegorz Sumara, PhD

Reviewer #1

“In this paper, the authors aim to identify downstream targets and signaling pathways of the serine/threonine kinase PKD3 in primary hepatocytes in vitro and in vivo. The PKD family comprises three isoforms (PKD1-3), the functions of which depend on the cell type and external signal cues. While the functions of the kinases has been extensively studied in vitro, knowledge of their physiological role is still limited. This paper thus addresses an important issue in the field. The authors employ two different phosphoproteomic approaches in which they compare primary PKD3 co-hepatocytes transduced with either adenovirus encoding EGFP or myc-labeled active PKD3. Using two different antibodies specific for phosphorylated PKD substrate motifs, several proteins were identified that showed increased phosphorylation at potential PKD phosphorylation sites in caPKD3-expressing hepatocytes compared to control cells. Among these, phenylalanine hydroxylase (PAH) appeared to be the most promising target. In agreement with this, the authors show that PKD3 is involved in the conversion of phenylalanine to tyrosine in vitro and in vivo.

The paper sheds light on a possible new function of PKD3 in liver metabolism and adds important information on isoform-specific functions of this member of the PKD family. While the data showing the involvement of PKD3 in tyrosine and phenylalanine metabolism in vitro and in vivo are strong and convincing, the experiments on PKD3-dependent phosphorylation of PAH and its functional connection with the metabolic phenotype of PKD3 co-mice are inadequate.”

We thank reviewer #1 for the enthusiastic assessment of our manuscript and constructive comments. His/her input modified the major focus of our manuscript and let us discover new aspects of PKD3-dependent signaling in hepatocytes.

“Major points need to be addressed here before the paper can be considered for publication.”

“Figure 3: As the authors point out, PKA has been identified as an upstream kinase of serine 16 in PAH in previous publications and phosphorylation of this site increases the affinity of PAH for its substrate Phe. Consequently, as serine 16-phosphorylated PAH was pulled down with a PKD pMOTIF substrate antibody the authors conclude that PAH could be a substrate for PKD3 as well. Unfortunately, this is not being pursued consistently by the authors.

Thus, in order to link PAH activity to PKD3 kinase activity and to exclude or prove that serine 16 in PAH is a direct PKD3 target, an in vitro kinase assay must be performed. If proven that PKD3 phosphorylates PAH directly, some discussion on whether the two kinases, PKD3 and PKA, compete for PAH phosphorylation and how this might be regulated would be expected. Also, in case PAH is not a direct target of PKD3 it would be interesting to have some thoughts about how PKA and PKD3 signaling might be linked.”

As suggested by reviewer # 1 we have tested if PKD3 phosphorylates PAH in the *in vitro* kinase assay. PKD3 did not phosphorylate PAH directly like PKA (Fig. 5A). These data prompt us to raise the hypothesis that PKD3 might affect PKA activity which leads to the increased phosphorylation of PAH. As revealed using an antibody raised against phosphorylation motive targeted by PKA (RRX*S/T) and antibody against active catalytic subunit of PKA (p-PKA C T197) in mice treated with PKD inhibitor or overexpressing its active form of PKD3, PKA activity is promoted by PKD3 (Fig. 5B-D).

Since PKA-dependent signaling has broader implications in hepatic physiology, we have decided to perform an extra set of experiments in mice and hepatocytes deficient for PKD or receiving inhibitor of this kinase. In line with the role of PKA in the regulation of hepatic metabolism, we figured out that PKD3 promotes glucose levels during fasting (Fig. 5D and E). These unexpected results in combination with our previous data have several implications. First of all, PKD3 co-stimulates PKA activity in response to glucagon. Additionally, PKD3 promotes fasting glucose levels. Finally, our proteomic data might reveal not only direct targets of PKD3 but also proteins that were phosphorylated by kinases that were activated or suppressed by PKD3. All of these possibilities are discussed in our manuscript.

“Figure 4: The authors propose a PKD3-dependent regulation of serine 16 phosphorylation in PAH. This assumption is based on a mobility shift of PAH upon PKD3ca expression. However, whether this mobility shift is related to serine 16 phosphorylation remains unclear. The PKD-pMOTIF substrate antibody that pulled down PAH should be used to demonstrate enhanced phosphorylation of endogenous PAH upon PKD3 expression in PKD3 co-cells in a Western blot analysis. This should be accompanied by an experiment that proves that this

antibody detects the phosphorylation of PAH on serine 16. These experiments would also help to functionally link the phosphorylation and activity of PAH to PKD3.”

We agree with reviewer #1 that such an experiment would directly prove that endogenous PAH would be phosphorylated in a PKD3-dependent manner. Unfortunately, directed against PAH does not work for immunoprecipitation in our hands. Therefore, for technical reasons, we could not perform this experiment. In addition, the in-vitro phosphorylation assay suggests that PAH is not a direct target of PKD3

“Figure 5: The authors state that glucagon promotes PKD3 activation regardless of the AA composition. However, this is difficult to understand from the Western blot shown. The PKD3-phospho-Blot shows two bands at the expected size, but which of them corresponds to the active PKD3? To clarify, the authors should compare with lysates obtained from PKD3-Ko-cells. Furthermore, if it is the lower band, there are significant differences in the amount of detectable phosphorylated PKD3, which seem to be dependent on the AA composition. In particular, the exclusive Phe / Tyr condition seems to have a negative influence on the glucagon-induced PKD3 activation. A quantitative analysis of three independent experiments is required to better evaluate the data. In addition, the total PKD3 values must be displayed. In addition, it would be important to show whether glucagon can promote the phosphorylation of PAH and the dephosphorylation of S6K and 4E-BP1 in PKD3 ko-hepatocytes. This would complement the data showing that the glucagon-induced increase in tyrosine levels is dependent on PKD3.”

To clarify this aspect, we carried out an in vivo experiment in mice injected with glucagon for 0, 5, 20 minutes. Glucagon increases the phosphorylation of PKD3 on ser744/748. This is abrogated in the liver isolated from PKD3-deficient mice injected with glucagon for 20 minutes (Fig. 4F). Interestingly, the abundance of PKD3 is also elevated after glucagon injection (Fig. 4F)

“Minor points:

Molecular weight markers on the blot in figure 4A are missing.”

We have fixed it.

Reviewer #2 (Comments to the Authors (Required)):

“The manuscript by Mayer et al. sheds new light on the action of PKD3 in the liver. The authors identify several putative targets of this kinase, and have further investigated on PAH and amino acid metabolism in response to glucagon stimulation.

Supporting data are: the PAH phospho-protein was found in 2 independent phosphoproteomics experiments; PKD motifs were previously identified in the PAH sequence; tyrosine levels are increased in PKD3 overexpressing mouse hepatocytes in vivo and in primary cultures; glucagon promotes activation of PKD3; primary hepatocytes lacking PKD3 fail to accumulate tyrosine after glucagon stimulation.

Overall, the paper is well written and the direct link between PKD3 and amino acid metabolism is supported by data.”

We thank reviewer #2 for the enthusiastic assessment of our manuscript.

“However, more evidence to support the glucagon-PKD3-PAH axis would be needed before publication:

• In Fig. 4B and C, the authors show tyrosine accumulation in hepatocytes and whole liver in PKD3 overexpression systems. Would that be still observed upon concomitant silencing of PAH?”

To test if PKD3 promotes tyrosine levels in hepatocytes in the PAH-dependent manner, we have inhibited PAH in cells overexpressing PKD3. By using a PAH inhibitor (panobinostat) which decreased the PKD3 induced tyrosine levels (Fig. 4E).

“Would a PKD3 inhibitor affect the glucagon induced upshift of PAH and tyrosine levels (related to Fig.5A)?”

As anticipated by reviewer #2 inhibition of PKDs using CRT0066101 ameliorated glucagon-induced tyrosine levels in hepatocytes (Figure 4H).

“• Primary PKD3-KO hepatocytes fail to increase tyrosine levels upon glucagon stimulation (Fig.5B); is this phenotype also seen in vivo, after glucagon stimulation or in fasting conditions?”

To find out if inhibition of PKDs in vivo also affects tyrosine levels we have challenged mice, which were previously treated with CRT0066101 inhibitor, with glucagon. However, in these animals, tyrosine levels were not different compare to the animals treated with the control solution (Figure 5I). Following the suggestion of another reviewer, we have demonstrated that PKD3 does not regulate PAH directly, but rather promotes the activity of PKA, which is a well-established regulator of PAH and hepatic metabolism in general (especially glucose metabolism). This prompted us to measure glucose levels in response to glucagon challenge and upon fasting in mice with the diminished activity of PKD3. Mice treated with CRT0066101 inhibitor presented markedly lower glucose levels upon short-term fasting (6h) and challenge with glucagon (Figure 5E). Of note also PKD3-deficient mice presented lower glucose levels upon prolonged fasting (Figure 5F). Suggesting, that not only inhibition of PKDs was effective but also that PKD3 presents a much broader function in the regulation of hepatic metabolism as predicted by changes in PKA activity. However, these new results raise further questions about the direct and indirect effects observed in our proteomic screens. We have discussed these issues in the new version of our manuscript.

“Minor comment:

• Please provide the metabolomics data. Supplementary Table 2 refers to phosphoproteomics and Supplementary Table 3 is not included, actually.”

We have corrected this omission.

Reviewer #3 (Comments to the Authors (Required)):

“Mayer et al have applied proteomic approaches to determine PKD3 targets in hepatocytes using a phospho-motif antibody pull-down approach coupled with mass spectrometry.

The authors used adenoviral vectors to express either GFP (control) or PKD3ca (a constitutively active PKD3 mutant) in PKD3 KO hepatocytes before immunoprecipitating cellular proteins that are reactive with either LxRxx(pS/pT) or with Rxx(pS/pT) motif antibodies. Subsequent mass spec analysis identified 310 proteins that were enriched in the immunoprecipitates, and approximately half of these contained at least one putative PKD substrate motif, [L/V/I]xRxx[pS/pT]. Of these, 24 proteins were identified in both screens and were thus identified by the authors as putative PKD3 targets in murine hepatocytes.

The authors next focused on one putative PKD3 target that was identified by their IP/mass spec approach, phenylalanine hydroxylase (PAH), concluding that PKD3 is involved in controlling phenylalanine conversion to tyrosine in primary hepatocytes. PAH contains two putative PKD phosphorylation sites, one of which surrounds Ser16, a known regulatory site that influences the affinity of PAH for phenylalanine and which has previously been shown to be phosphorylated in response to glucagon in a PKA-dependent manner. In this manuscript the authors show that expression of PKD3ca in PKD3 KO cells was sufficient to induce a MW shift in the PAH protein, indicative of a post-translational modification event. The authors next demonstrated, using an in vitro conversion assay, that PKD3ca expression boosted cellular tyrosine levels in PKD3 KO cells (compared to control GFP+ cells) under Tyr starvation conditions, independent of whether extracellular Phe was present or not. A similar result was observed for basal levels of tyrosine in the livers of PKD3ca transgenic mice versus control mice.

Finally, the authors investigated whether PKD3 regulates glucagon-induced phenylalanine metabolism, showing that glucagon treatment stimulates the conversion of Phe to Tyr in wild-type hepatocytes but not in PKD3 KO hepatocytes under conditions of Phe/Tyr starvation. This correlates with findings showing that a large MW shift in PAH is induced by glucagon in Phe/Tyr-starved cells (and that PKD3 T-loop phosphorylation is also increased by glucagon, albeit in an AA-independent manner).

This is a very nice study that advances the field in terms of how PKD3 is involved in glucagon signaling/cellular metabolism, as well as providing a great resource for further investigations. If the authors can address the issues highlighted below (by either including additional supporting data or by modifying their results analysis/discussion) then I recommend publication.”

We thank reviewer #3 for his/her assessment of our manuscript.

“1.Expression of a constitutively active kinase can cause spurious results, due to

dysregulated activity (spatial and/or temporal) - i.e. just because a constitutively active mutant can do something, does it really regulate that event in vivo? The authors do include key data in Figure 5 showing that deletion of PKD3 negatively impacts on Phe to Tyr conversion in cultured hepatocytes compared to wild-type cells. This data might be better placed earlier in the manuscript however to really highlight the essential role of PKD3 in this metabolic process.”

Following the suggestion of reviewer #3, we have exposed the data originally presented in Figure 5A, earlier.

“2.Inclusion of additional data would strengthen the main message of this manuscript. e.g Is the PAH MW shift reduced in PKD3 KO cells vs wild-type cells? What effect do PKD specific inhibitors (and/or expression of a catalytically dead PKD3 mutant) have on glucagon-induced PAH MW shifts? Are tyrosine levels reduced in the livers or PKD3 KO vs wild-type mice under different feeding conditions?”

Following the suggestions of reviewer #3, we have tested the impact of PKD inhibitor on levels of tyrosine in isolated hepatocytes and livers of mice. While inhibition of PKDs in isolated hepatocytes resulted in amelioration of glucagon-induced tyrosine levels (Figure 4H), in mice PKD inhibitor did not affect tyrosine content in livers (Figure 4I). We have discussed this discrepancy in our manuscript.

“3.The authors conclusion that glucagon regulates PKD3 T loop phosphorylation (Figure 5A) is not convincing as the identity of PKD3 amongst the protein species present is not clear (several do not appear to be glucagon regulated from the data shown) - cells from PKD3 KO mice could be used as a control perhaps?”

Following the suggestion of reviewer #3, we have replaced the WB presented originally in figure 5A, with the WB performed on livers of wild-type and PKD3-deficient mice stimulated with glucagon (current Fig. 4F). These data confirm that PKD3 is activated in the liver by glucagon. Moreover, our new data indicate that PKD3 is more broadly implicated in the regulation of hepatic metabolism and regulates classical downstream pathways induced by glucagon stimulation, namely PKA (Figure 5B-D). This prompt us to also test if PKD3 regulates glucagon-induced or fasting glucose levels. In fact, in mice treated with PKD inhibitor, glucagon-induce glucose levels were significantly lower than in corresponding control animals (Fig. 5D). Also, the fasting glucose levels in PKD3-deficient mice were significantly reduced (Fig. 5E).

“4.How reproducible are the three biological replicates for the IP/mass spec experiments shown in Figures 1 and 2? This information should be provided.”

We have included this information in the materials section.

“5.The authors should introduce a note of caution when interpreting/discussing the results shown in Figures 1-3, with regard to the potential for non-specific interactions with the two phospho-motif antibodies used. e.g. Protein kinases other than PKDs can phosphorylate LxRxx(pS/pT) or Rxx(pS/pT) motifs, also some of these interactions may be occurring in a

non-phospho-specific manner and thus could be influenced by changes in protein expression level.”

We fully agree with reviewer #3 that many of the observed putative targets of PKD3 might be identified due to the unspecific interactions or could be like in the case of PAH phosphorylated by other kinases which are induced/suppressed by manipulation of PKD3. We have included this caution in the section of the discussion.

“6.As shown in Figure 4B, tyrosine levels increase in control (GFP+) PKD3 KO hepatocytes when extracellular Phe levels are increased, suggesting that PKD3 may not be the sole regulator of PAH and/or Phe to Tyr conversion?”

PAH activity is regulated in an allosteric manner and by phosphorylation provided by PKA. Our new data suggest that PKD3 regulates PAH activity by targeting PKA. We have discussed this in our manuscript.

“7.The number of replicate experiments and specific details on the nature and results of any statistical tests, including specific p values, should be indicated for all experiments in the manuscript.”

We have provided this information in the figure legends.

April 7, 2021

Re: Life Science Alliance manuscript #LSA-2020-00863-TR

Dr. Grzegorz Sumara
Rudolf Virchow Center for Experimental Biomedicine University of Würzburg
Josef-Schneider-Straße 2, Haus D15
Würzburg 97080
Germany

Dear Dr. Sumara,

Thank you for submitting your revised manuscript entitled "A phosphoproteomic approach reveals that PKD3 controls PKA-mediated glucose and tyrosine metabolism" to Life Science Alliance. The manuscript has been seen by the original reviewers whose comments are appended below. While the reviewers continue to be overall positive about the work in terms of its suitability for Life Science Alliance, some important issues remain.

We apologize for this unusual and extended delay in getting back to you. As you will note from the reviewers' comments, R1 has raised some important concerns about the new data that has been included in the revised manuscript. Thus, while our general policy is that papers are considered through only one revision cycle, given that the suggested changes are relatively minor and pertaining to the new data added in revision, we are open to one additional short round of revision.

Please submit the final revision within one month, along with a letter that includes a point by point response to the remaining reviewer comments.

-- Summary blurb (enter in submission system): A short text summarizing in a single sentence the study (max. 200 characters including spaces). This text is used in conjunction with the titles of papers, hence should be informative and complementary to the title and running title. It should

describe the context and significance of the findings for a general readership; it should be written in the present tense and refer to the work in the third person. Author names should not be mentioned.

B. MANUSCRIPT ORGANIZATION AND FORMATTING:

Sincerely,

Shachi Bhatt, Ph.D.
Executive Editor
Life Science Alliance
<http://www.lsjournal.org>
Tweet @SciBhatt @LSAJournal

Reviewer #1 (Comments to the Authors (Required)):

The authors have put a lot of effort in the revision of this manuscript and have also addressed most of my major points. From their data, it is now clear that PKD3 is involved in glucose and tyrosine metabolism in the liver, presumably by regulating PKA activity. However, because the authors added new data on a possible connection between PKD3 and PKA, this raises questions that should be addressed to strengthen their conclusions.

Major point:

Figure 5: This figure concentrates on the potential link between PKD3 and PKA. From the experiments shown, it is clear that overexpression of caPKD3 and inhibition of PKD3 increases and decreases the detection of proteins with the PKA substrate antibody, respectively. However, since the motif of the PKA substrate antibody overlaps with that of the PKD substrate antibody, it is not clear whether only PKA-specific events are exclusively detected. In addition, CREB was identified as a PKD substrate, so its phosphorylation at S133 cannot be used as a reliable indicator of PKA activity (Johanessen et al., 2007). Data showing that the PKD-specific inhibitor has an inhibitory effect on PKA phosphorylation are convincing, but can the authors rule out that it is not an off-target effect of CRT? It would be important to support these data by an experiment in which they show that in PKD3-KO mice (or primary hepatocytes) challenged with glucagon, PKA-Thr phosphorylation and/or PKA substrate antibody detection cannot be decreased by CRT treatment. This would also reinforce the data shown in 5E.

The statistical method used to analyze the data in 5 C and D is not stated - I guess a student's t test has been used as two groups are compared? In this regard, the strong significant difference shown for CRT and DMSO with respect to PKA-Thr phosphorylation is somewhat curious because the error bars are quite large. Can the authors confirm this?

Minor point:

Figure 4F: Quantification of the blots shows that PKD3 activity increases with 20 min of glucagon stimulation; however, PKD3 levels also increase to the same extent. This is surprising because PKD proteins are thought to be relatively stable and have a long half-life (approximately 24 h). Nevertheless, looking at the data presented it is likely that the phospho-PKD3/PKD3 ratio, and thus

PKD3 activity, remains unchanged with glucagon stimulation. The authors should thus reconsider their statement in this regard.

The PKD3 blot shows two bands in WT animals, both absent in PKD3 ko animals, but which one corresponds to PKD3? Assuming the molecular weight of PKD3, it must be the upper band, but this should be indicated.

Reviewer #2 (Comments to the Authors (Required)):

In this revised version, the authors have addressed my concerns. I have no further objections and recommend publication in LSA.

Just one minor detail, the arrangement of the individual panels within the figures could be improved, especially with regards to their proportions and font sizes.

Dear Dr. Bhatt,

Thank you very much for all your efforts to improve our manuscript.

In our opinion, we have addressed all of the comments of both reviewers and modified our manuscript accordingly. We have also updated the induration to consider new relevant publications in the field.

Below you can find a point-by-point response to the reviewers' concerns.

Thank you once again for your help.

Best regards,

Grzegorz Sumara, PhD

Response to the reviewers' comments:

Reviewer #1

"The authors have put a lot of effort in the revision of this manuscript and have also addressed most of my major points. From their data, it is now clear that PKD3 is involved in glucose and tyrosine metabolism in the liver, presumably by regulating PKA activity. However, because the authors added new data on a possible connection between PKD3 and PKA, this raises questions that should be addressed to strengthen their conclusions."

We thank reviewer #1 for all his/her efforts to improve our manuscript.

"Major point:"

"Figure 5: This figure concentrates on the potential link between PKD3 and PKA. From the experiments shown, it is clear that overexpression of caPKD3 and inhibition of PKD3 increases and decreases the detection of proteins with the PKA substrate antibody, respectively. However, since the motif of the PKA substrate antibody overlaps with that of the PKD substrate antibody, it is not clear whether only PKA-specific events are exclusively detected. In addition, CREB was identified as a PKD substrate, so its phosphorylation at S133 cannot be used as a reliable indicator of PKA activity (Johanessen et al., 2007). Data showing that the PKD-specific inhibitor has an inhibitory effect on PKA phosphorylation are convincing, but can the authors rule out that it is not an off-target effect of CRT? It would be important to support these data by an experiment in which they show that in PKD3-KO mice (or primary hepatocytes) challenged with glucagon, PKA-Thr phosphorylation and/or PKA substrate antibody detection cannot be decreased by CRT treatment. This would also reinforce the data shown in 5E."

We agree with reviewer #1 that PKD inhibitor, CRT0066101, might not be fully specific. Therefore, we have examined if silencing of PKD3 in primary hepatocytes using shRNA, also results in diminished PKA activity. Silencing of PKD3 resulted in a significant reduction of PKA activity, in cells depleted of PKD3, and CRT0066101 did not significantly decrease PKA activation in the absence of PKD3 (Fig. 5E and F).

"The statistical method used to analyze the data in 5 C and D is not stated - I guess a student's t-test has been used as two groups are compared? In this regard, the strong significant difference shown for CRT and DMSO with respect to PKA-Thr phosphorylation is somewhat curious because the error bars are quite large. Can the authors confirm this?"

In fact, we have made a mistake in the calculations, which have been fixed in the current version. We thank reviewer #1 for finding this error.

"Minor point:"

"Figure 4F: Quantification of the blots shows that PKD3 activity increases with 20 min of glucagon stimulation; however, PKD3 levels also increase to the same extent. This is surprising because PKD proteins are thought to be relatively stable and have a long half-life (approximately 24 h). Nevertheless, looking at the data presented it is likely that the phospho-

PKD3/PKD3 ratio, and thus PKD3 activity, remains unchanged with glucagon stimulation. The authors should thus reconsider their statement in this regard.”

We agree with reviewer #1, we have changed the sentence “*Of note, glucagon increased PKD activity in the livers of mice and also to a certain degree abundance of PKD3 (Fig. 4F).*” to “*Of note, glucagon increased the abundance of active PKD and also PKD3 in the liver (Fig. 4F).*”

“The PKD3 blot shows two bands in WT animals, both absent in PKD3 ko animals, but which one corresponds to PKD3? Assuming the molecular weight of PKD3, it must be the upper band, but this should be indicated.”

We have indicated in the figure legend that the upper band corresponds to the predicted molecular weight of PKD3.

Reviewer #2

“In this revised version, the authors have addressed my concerns. I have no further objections and recommend publication in LSA.”

We thank reviewer #2 for his/her impact on our manuscript.

“Just one minor detail, the arrangement of the individual panels within the figures could be improved, especially with regards to their proportions and font sizes.”

We have fixed it.

May 19, 2021

RE: Life Science Alliance Manuscript #LSA-2020-00863-TRR

Dr. Grzegorz Sumara
Rudolf Virchow Center for Experimental Biomedicine University of Würzburg
Josef-Schneider-Straße 2, Haus D15
Würzburg 97080
Germany

Dear Dr. Sumara,

Thank you for submitting your revised manuscript entitled "A phosphoproteomic approach reveals that PKD3 controls PKA-mediated glucose and tyrosine metabolism". We would be happy to publish your paper in Life Science Alliance pending final revisions necessary to meet our formatting guidelines.

Along with the points listed below, please also attend to the following:

- please add ORCID ID for the corresponding author-you should have received instructions on how to do so
- please label the Summary as an "Abstract"
- please add a conflict of interest statement to your main manuscript text
- please add your table legends to the main manuscript text after the main figure legends
- please provide higher quality images for all blots shown in Figure 5
- please deposit the large datasets from this manuscript in any of the publicly available depositories and provide the accession number under 'Data Availability' section in the manuscript; in compliance with our editorial policies (<https://www.life-science-alliance.org/manuscript-prep#datadepot>)

A. FINAL FILES:

-- High-resolution figure, supplementary figure and video files uploaded as individual files: See our detailed guidelines for preparing your production-ready images, <https://www.life-science->

alliance.org/authors

B. MANUSCRIPT ORGANIZATION AND FORMATTING:

Sincerely,

Shachi Bhatt, Ph.D.
Executive Editor
Life Science Alliance
<http://www.lsjournal.org>
Tweet @SciBhatt @LSAJournal

Reviewer #1 (Comments to the Authors (Required)):

The authors have addressed my concerns, I have no further objections and recommend publication.

June 10, 2021

RE: Life Science Alliance Manuscript #LSA-2020-00863-TRRR

Dr. Grzegorz Sumara
Rudolf Virchow Center for Experimental Biomedicine University of Würzburg
Josef-Schneider-Straße 2, Haus D15
Würzburg 97080
Germany

Dear Dr. Sumara,

Thank you for submitting your Research Article entitled "A phosphoproteomic approach reveals that PKD3 controls PKA-mediated glucose and tyrosine metabolism". It is a pleasure to let you know that your manuscript is now accepted for publication in Life Science Alliance. Congratulations on this interesting work.

DISTRIBUTION OF MATERIALS:

Again, congratulations on a very nice paper. I hope you found the review process to be constructive and are pleased with how the manuscript was handled editorially. We look forward to future exciting submissions from your lab.

Sincerely,
